# Machine learning-assisted fluoroscopy of bladder function in awake mice

Helene De Bruyn[1,2], Nikky Corthout[3], Sebastian Munck[3], Wouter Everaerts[4], Thomas Voets[1,2]*

[1]Laboratory of Ion Channel Research (LICR), VIB-KU Leuven Center for Brain & Disease Research, Leuven, Belgium; [2]Department of Cellular and Molecular Medicine, KU Leuven, Leuven, Belgium; [3]VIB BioImaging Core – VIB-KU Leuven Center for Brain & Disease Research, KU Leuven Neuroscience Department, Leuven, Belgium; [4]Laboratory of Organ System, Department of Development and Regeneration, KU Leuven, Leuven, Belgium

**Abstract** Understanding the lower urinary tract (LUT) and development of highly needed novel therapies to treat LUT disorders depends on accurate techniques to monitor LUT (dys)function in preclinical models. We recently developed videocystometry in rodents, which combines intravesical pressure measurements with X-ray-based fluoroscopy of the LUT, allowing the in vivo analysis of the process of urine storage and voiding with unprecedented detail. Videocystometry relies on the precise contrast-based determination of the bladder volume at high temporal resolution, which can readily be achieved in anesthetized or otherwise motion-restricted mice but not in awake and freely moving animals. To overcome this limitation, we developed a machine-learning method, in which we trained a neural network to automatically detect the bladder in fluoroscopic images, allowing the automatic analysis of bladder filling and voiding cycles based on large sets of time-lapse fluoroscopic images (>3 hr at 30 images/s) from behaving mice and in a noninvasive manner. With this approach, we found that urethane, an injectable anesthetic that is commonly used in preclinical urological research, has a profound, dose-dependent effect on urethral relaxation and voiding duration. Moreover, both in awake and in anesthetized mice, the bladder capacity was decreased ~fourfold when cystometry was performed acutely after surgical implantation of a suprapubic catheter. Our findings provide a paradigm for the noninvasive, in vivo monitoring of a hollow organ in behaving animals and pinpoint important limitations of the current gold standard techniques to study the LUT in mice.

*For correspondence:
thomas.voets@kuleuven.be

Competing interest: The authors declare that no competing interests exist.

## Editor's evaluation

The described technique allows for accurate and detailed measurement of intravesical pressures, urethral flow rate, inter-contraction intervals, and residual bladder volume without the need for catheter implantation, and in anesthetized and awake mice. This method has the potential for broad application ranging from the elucidation of molecular mechanisms of bladder control to drug development for LUT dysfunction. The reporting on the AI training process, a user manual and the trained neural network makes this machine-learning approach to image-analysis a useful tool for LUT studies in other labs.

## Introduction

The lower urinary tract (LUT), consisting of the bladder and urethra, plays an important role in our daily life functioning. It is responsible for storing urine and emptying the bladder at an appropriate

**eLife digest** Healthy adults empty their bladder many times a day with little thought. This seemingly simple process requires communication between the lower urinary tract and the central nervous system. About one in five adults experience conditions like urinary incontinence, urgency, or bladder pain caused by impairments in their lower urinary tract. Despite the harmful effects these conditions have on people's health and well-being, few good treatments are available.

Mice are often used to study lower urinary tract conditions and treatments. One common technique is to fill a mouse's bladder using a catheter and measure changes in pressure as the bladder empties and refills. But these procedures and the anesthesia used during them may affect bladder function and skew results.

Here, De Bruyn et al. have developed a new technique that allows scientists to measure bladder function in awake, freely moving mice. The mice's bladders were photographed using a specialized X-ray based fluoroscope that captured 30 images per second over the course of three hours. A machine learning algorithm was then applied which can automatically detect the circumference of the bladder in each captured image (over 30,000 in total) and quantify its volume. This makes it is possible to measure the bladder as it empties and fills even if the mice move between time frames.

The new approach showed that 'gold standard' commonly used methods have a profound effect on the bladder. Surgical implantation of a catheter reduced the bladder to a quarter of its capacity. In addition, one of the most widely used anesthetic drugs in urinary tract research was found to affect the bladder's ability to drain.

The technique created by De Bruyn et al. provides a new way to study lower urinary tract function and disease in awake, moving animals. This tool would be easy for other academic and pharmaceutical laboratories to implement, and may help scientists discover new therapies for lower urinary tract conditions.

time and place. Proper LUT functioning is the result of complex communication pathways between the LUT and central nervous system. An estimated 20% of the population suffers from some sort of lower urinary tract dysfunction (LUTd), such as urinary incontinence, frequency, urgency, or bladder pain, which can have a strongly negative impact on health and wellbeing. Current treatments for LUTd generally lack efficacy and often come with bothersome side effects. Therefore, there is a high need for new treatment options, which depend on adequate preclinical models to study mechanisms of LUT control and to test potential novel therapies.

Due to their anatomical and physiological similarities to humans, their cost-effectiveness, and the availability of a vast number of genetically modified strains, mice are favored models in preclinical LUT research (*Ito et al., 2017*). In the last decades, cystometry has been considered the gold standard technique to study bladder function in mice in vivo. Cystometry is an experimental procedure whereby the bladder is continuously filled via an implanted catheter, and pressure inside the bladder is monitored during multiple filling and voiding cycles. More recently, this technique has sometimes been complemented with urethral electromyography (UEM) to provide simultaneous information regarding the activity of the urethral sphincters (*Kadekawa et al., 2016*). Whereas these approaches have fueled important advances in our understanding of the cellular and molecular mechanisms that steer the LUT, they also come with important drawbacks. First, cystometry and UEM do not provide quantitative information regarding the filling state of the bladder or the urethral flow, whereas changes in these parameters are hallmarks of various LUTds. Second, cystometry and UEM require the introduction of, respectively, a catheter in the bladder wall and metal electrodes in the urethral sphincter, both of which can affect LUT function. Third, cystometry and UEM are mostly performed on urethane anesthetized or physically restrained mice to avoid complications and artifacts due to animal movement. The consequences of these procedures on LUT function in mice are incompletely understood.

To overcome some of these limitations, we recently introduced videocystometry, a new approach combining cystometry with continuous, fluoroscopy-based bladder imaging. By imaging the contrast-filled bladder at video rate, this approach provides quantitative and time-resolved measurements of bladder volume and urethral flow, along with other relevant aspects of LUT function such as vesicoureteral reflux and micromotions of the bladder wall (*Franken et al., 2021*). Notably, the accurate analysis

of bladder volume and urethral flow during consecutive filling–voiding cycles relies on the precise delineation of the bladder. This is readily achieved in anesthetized or otherwise motion-restrained mice, where the constant background of fluoroscopic images can be subtracted. However, in awake, unrestrained, and behaving animals, the position of the bladder relative to other structures visible in the fluoroscopic images (e.g., bones) changes constantly. Therefore, our earlier analyses relied on manual tracking of the border of the bladder, which is feasible for a limited number of fluoroscopic images but clearly unworkable for typical minutes-to-hours-long imaging sessions at 30 images/s, yielding >$10^5$ images.

To address this problem, we developed a machine-learning approach where we trained a neural network that automatically and faithfully detects the bladder in large sets of time-lapse fluoroscopic images in mice, allowing accurate determination of volume- and flow-related parameters in behaving mice. The combination of videocystometry and machine learning allowed us to pinpoint the profound effects of urethane, an injection anesthetic standardly used in preclinical bladder research, on the function of the LUT. Furthermore, we provide proof of principle for the use of fluoroscopy combined with machine-learning-based image analysis to noninvasively monitor bladder function in awake animals, which reveals the significant impact of suprapubic catheterization on functional bladder capacity (BC).

## Results

### Machine-learning-based analysis of fluoroscopic images of the LUT

We combined classic cystometry with fluoroscopy-based imaging to study LUT function in awake, unrestrained mice. In previous work, we had established two methods to determine bladder volume ($V_{ves}$) from fluoroscopic images. The first method, which is based on the linear relation between background-corrected opacity and volume of contrast-filled bladders, is only applicable to recordings with a stable background, that is in anesthetized or otherwise immobile animals. The second method, which is based on the approximation of the bladder as a spheroid, depends on the delineation of the bladder wall but does not require background subtraction, making it, in principle, suitable to determine bladder volume from images of moving mice. However, considering that our videocystometry experiments lasted up to 3 hr, at a sampling rate of 30 images/s, manual delineation of the bladder wall in these large sets of time-lapse fluoroscopic images was impossible. We therefore developed a machine-learning protocol to automatically identify the bladder border in fluoroscopy images, independent of the position or posture of the mouse or the degree of bladder filling. To achieve this, we performed several cycles of training of a neural network using a set of human-annotated fluoroscopy images (*Figure 1A*).

The machine-learning protocol was able successfully identify the bladder border compared to manual bladder wall delineation in time-lapse image sequences, irrespective of the posture of the animal or phase of the filling/voiding cycle (*Figure 1B–C*; *Figure 1—video 1*, and *Figure 1—video 2*), with the exception of brief failures of one or two frames during abrupt mouse movements or when the bladder was almost fully emptied. During efficient voids, the lowest volume at which the bladder was successfully detected by the machine-learning protocol was 4.1 ± 2.2 µl (*n* = 15 voids from 3 animals). We calculated the Dice similarity index (DISI; see Materials and methods) to benchmark the performance of the neural network (*Dice, 1945*). This analysis yielded a median DISI value of 0.95 (mean ± standard deviation [SD]: 0.92 ± 0.13), indicating overall excellent agreement between human annotation and network prediction. We therefore used this approach to determine time-dependent changes in bladder volume based on the spheroid approximation (*Figure 1B*), along with other volume-related voiding parameters: BC, residual volume (RV), and voiding efficiency ($E_{voiding}$), as described previously (*Franken et al., 2021*).

### Dose-dependent effect of urethane anesthesia on voiding function

Next, we used this approach to evaluate the dose-dependent effects of urethane, the most predominantly used anesthetic during in vivo cystometry in rodents, on LUT function. Urethane was administered at four time points, leading to a stepwise increase of the total dose from 0 to 1.2 g/kg bodyweight, the latter being the typical dose used in most rodent studies. This protocol caused the animals to gradually transit from a nonanesthetized state, where they could freely move within the boundaries of the recording box, to a fully anesthetized, largely immobile state. Before dosing and

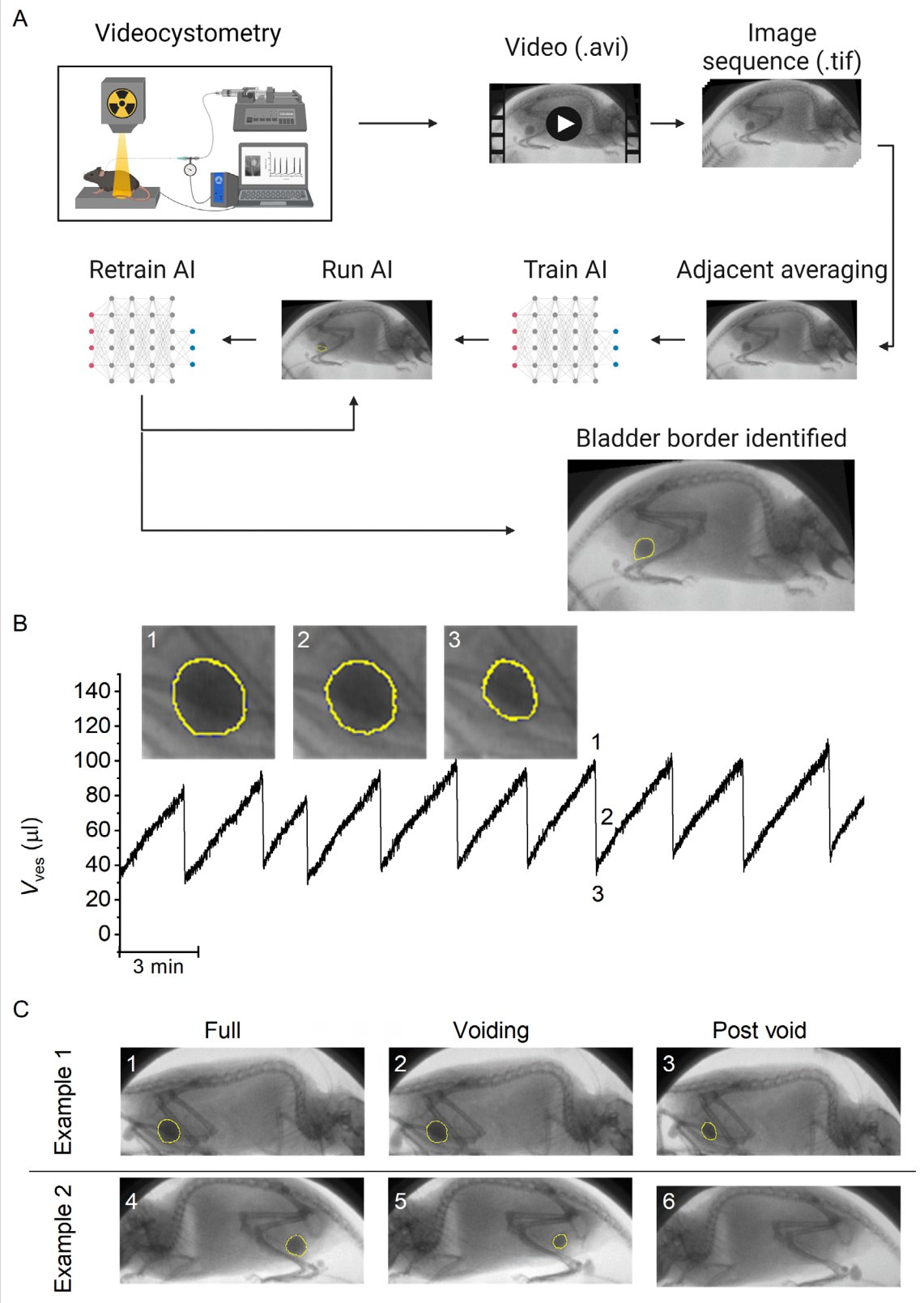

**Figure 1.** A machine-learning protocol for the automated annotation of the bladder in fluoroscopic images. (**A**) Image analysis protocol using artificial intelligence-based automatic annotation of the bladder border. (**B**) Representative bladder volume trace from an anesthetized animal (1.2 g/kg urethane) based on artificial assisted automated annotation of the bladder. Annotated bladder images (blue line), corresponding to different

*Figure 1 continued on next page*

*Figure 1 continued*

filling states of the bladder, are shown. (**C**) Examples showing annotated bladders before, during, and after voiding in different positions. Example 1 represents the same animal as in panel B, whereas Example 2 represents an awake animal.

The online version of this article includes the following video and figure supplement(s) for figure 1:

**Figure supplement 1.** Effect of rolling averaging on voiding parameters.

**Figure 1—video 1.** Video showing fluoroscopic imaging of a void in an awake mouse.

https://elifesciences.org/articles/79378/figures#fig1video1

**Figure 1—video 2.** Video showing fluoroscopic imaging of a void in a urethane anesthetized mouse.

https://elifesciences.org/articles/79378/figures#fig1video2

after each urethane administration, videocystometry was performed during 40 min ($n$ = 5), resulting in 200-min-long recordings. These encompass approximately 360,000 time-lapse fluoroscopy images per experiment, from which we derived time-dependent changes in volume and the various volume-related parameters (*Figure 2A*). Urethane did not have a statistically significant effect on BC (*Figure 2C*), but exhibited a pronounced and dose-dependent effect on RV and $E_{voiding}$ (*Figure 2D and E*). Indeed, whereas awake animals completely emptied their bladder at each void, urethane at doses >0.3 g/kg caused a dose-dependent increase in the volume that remained after each void and a corresponding decrease in $E_{voiding}$ from ~100% to below 50% (*Figure 2E*). The intercontractile interval (ICI) decreased concomitantly with the reduced efficiency of voiding (*Figure 2F*). Note that animals that were immediately treated with a single 1.2 g/kg-dose yielded similar values for BC, RV, and $E_{voiding}$ as animals that reached this dose at the end of the stepwise dosing protocol, indicating that the reduced voiding efficiency and increased RV are due to the urethane rather than to fatigue (*Figure 2C–F*, blue) (*Franken et al., 2021*). Taken together, our findings indicate that urethane leads to a pronounced and dose-dependent reduction of the ability of the bladder to empty during voiding.

Next, we zoomed in on individual voids to identify changes in void duration or urethral flow that could explain the dose-dependent reduction in voiding efficiency by urethane (*Figure 3A*). Increasing doses of urethane resulted in a gradual decrease in void duration, as quantified by the 20–80% time interval ($t_{20-80}$) (*Figure 3C*). Urethane also affected the urethral flow rate (UFR), which we determined from the time derivative of the changes in bladder volume, showing a gradual decrease at increasing doses (*Figure 3D*). These findings demonstrate that incomplete bladder emptying at higher urethane doses is the result of both shorter void duration and reduced flow rate of urine expulsion from the bladder during a void. Analysis of the combined intravesical pressure and UFR measurements further revealed that the urethane-induced reduction of urethral flow was the result of a lower urethral flow conductance (UFC), while intravesical pressure during a void was not significantly affected (*Figure 3E*). Taken together, these findings reveal that urethane has a profound and dose-dependent effect on voiding efficiency, which can be attributed to a reduction in the extent and duration of urethral relaxation. Moreover, when we analyzed the 80 s period preceding each void, we found a dose-dependent decrease in the number of nonvoiding contractions (*Figure 3B and F*).

## Non-invasive fluoroscopic monitoring of the LUT

In classic cystometry and videocystometry experiments, surgery is performed to implant a catheter into the bladder dome, through which the bladder is filled at a constant, often supraphysiological filling rate. The impact of this invasive intervention on bladder function is poorly understood. We adapted our fluoroscopy-based approach to monitoring bladder volume in a noninvasive manner in awake, unrestrained animals. Injection of a bolus of contrast solution subcutaneously in the scruff region resulted in the excretion of iodine-containing urine, enabling fluoroscopy-based imaging of the bladder without catheter implantation. In this setting, the filling rate of the bladder is solely determined by the renal urine output, which we accelerated within a physiologically acceptable range using a diuretic. Animals were imaged during 120 min, during which voids occurred spontaneously at a rate of 1–2 voids per hour, and the machine-learning protocol was used to automatically detect the bladder border. In these experiments, we measured an average filling rate of 7.3 ± 2.8 µl/min (mean ± SD). This approach, fluoroscopic volumetry, allowed for the first time to accurately and continuously quantify volume- and flow-related parameters noninvasively in awake and behaving animals (*Figure 4A–C*), albeit without concomitant pressure measurement.

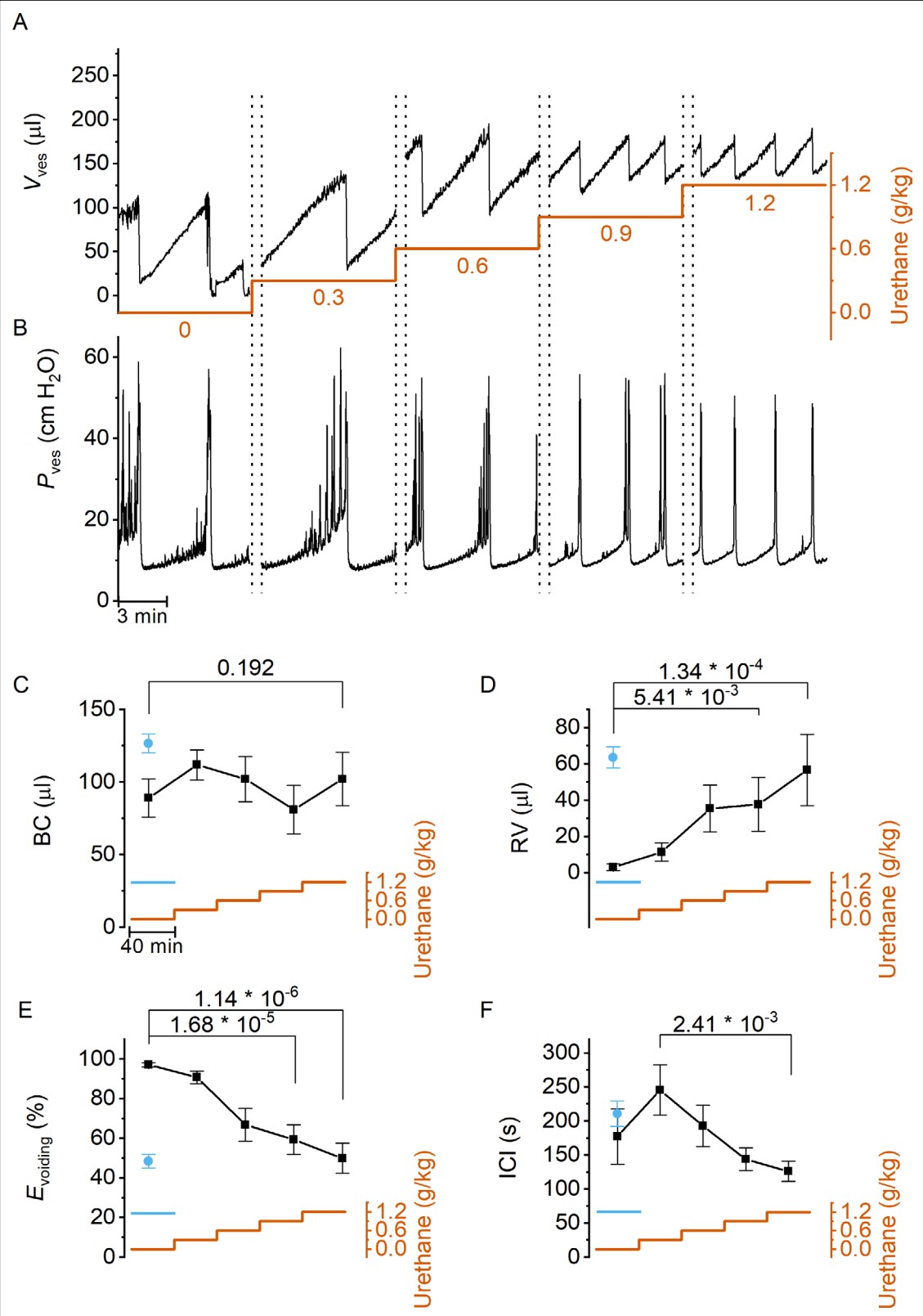

**Figure 2.** Dose-dependent effects of urethane on bladder volume and voiding efficiency. (**A, B**) Short excerpts of simultaneous bladder volume and intravesical pressure recordings traces at increasing cumulative doses of urethane in catheterized mice. (**C–F**) Average bladder capacity, residual volume, voiding efficiency, and intercontractile interval at increasing doses of urethane (mean ± standard error of the mean [SEM]). Animals were assayed for 40 min at each dose, after which the dose as increased as indicated by the orange trace. For comparison, data from mice that immediately received

*Figure 2 continued on next page*

*Figure 2 continued*

a single dose of 1.2 g/kg at the start of the recording (*Franken et al., 2021*) are also included, shown by the blue symbols and blue line. One-way repeated measures analysis of variance (ANOVA) was used to test for statistically significant differences with the 0.0 g/kg condition.

We performed fluoroscopic volumetry in four animals, first in the awake conditions and subsequently after administration of urethane at a dose of 1.2 g/kg. Notably, both in the awake and in the anesthetized conditions, we observed a dramatically larger BC in comparison with catheterized animals: BC amounted to 364 ± 60 µl (compared to 89 ± 13 µl in catheterized animals, p = 0.0015) in the absence of urethane, and to 380 ± 28 µl (compared to 102 ± 18 µl, p = 0.02) after dosing 1.2 g/kg urethane (*Figure 4D*). Like in the catheterized animals, we detected complete voiding in the awake animals, whereas anesthetized animals had substantial RV, corresponding to a voiding efficiency of ~60% (*Figure 4E–G*). In comparison with catheterized animals, both the maximal UFR and the void duration were significantly increased (*Figure 4H–I*). Notably, whereas urethane caused a shortening of the void duration in catheterized animals (*Figure 3*), a prolongation was observed in the nonoperated animals (*Figure 4H*). Taken together, these results demonstrate the feasibility of machine learning-assisted bladder fluoroscopy to study bladder function in a noninvasive manner and highlight important effects of catheter implantation on bladder filling and voiding.

## Discussion

The process of bladder filling and voiding is regulated by complex and incompletely understood signaling pathways between the central nervous system and the LUT, which coordinate the contractile state of the bladder muscle and urinary sphincters. Dysregulation of these processes can lead to a plethora of bothersome LUT disorders, for which current treatments are often unsatisfactory. Rodents are widely used in preclinical research aimed at understanding the (patho)physiology of the LUT and at the development of novel treatments. However, the current state-of-the-art methods and approaches to study bladder function in rodents have various drawbacks, which may affect interpretation and compromise translation of the findings to the human situation. For instance, the gold standard technique in the assessment of LUT function in rodents, cystometry, requires the surgical implantation of a catheter through the bladder wall – the consequences of this invasive technique on bladder sensitivity and storage capacity are poorly understood. Moreover, invasive techniques such as cystometry and UEM are generally performed on anesthetized or physically restrained mice, which may substantially affect LUT function. Existing noninvasive techniques in awake, freely moving rodents, which depend on the detection of voided urine on filter paper or balances, lack detailed temporal resolution and deliver only limited information on the voiding process. Finally, none of these techniques provides precise and continuous measurements of the bladder volume.

Recently, we introduced videocystometry, combining cystometry with high-speed fluoroscopy imaging of the contrast-filled bladder. We established that bladder volume and urethral flow can be accurately derived from fluoroscopic images, based either on image opacity after background correction or on precise annotation of the bladder border. In awake and nonrestrained animals, determination of bladder volume based on image opacity is not feasible due to the continuously changing background of the behaving animal. Manual annotation of the bladder border is unworkable for typical hours-long imaging sessions at 30 images per second, yielding several >300,000 images per experiment. In this study, we have overcome this limitation by developing and successfully implementing a machine-learning protocol, which allowed the automated identification of the bladder border and derived volumetric parameters from large sets of time-lapse images in awake, behaving mice.

Next, we used the machine learning-assisted fluoroscopy to quantify the effect of anesthesia on LUT function. Urethane, which can be administered subcutaneously, intraperitoneal, or intravenously, is the most commonly used anesthetic in LUT research, preferred above other anesthetics that are known to suppress the micturition reflex or that create a less stable anesthesia (*Field and Lang, 1988*; *Matsuura and Downie, 2000*). Despite its broad use in preclinical LUT research, the mechanism of action of urethane is not clearly understood, and the impact on LUT physiology remains poorly understood (*Matsuura and Downie, 2000*; *Maggi and Meli, 1986*; *Cannon and Damaser, 2001*). We performed videocystometry in mice implanted with a suprapubic catheter, where we started the recording in awake animals and monitored changes in bladder function upon stepwise increases in

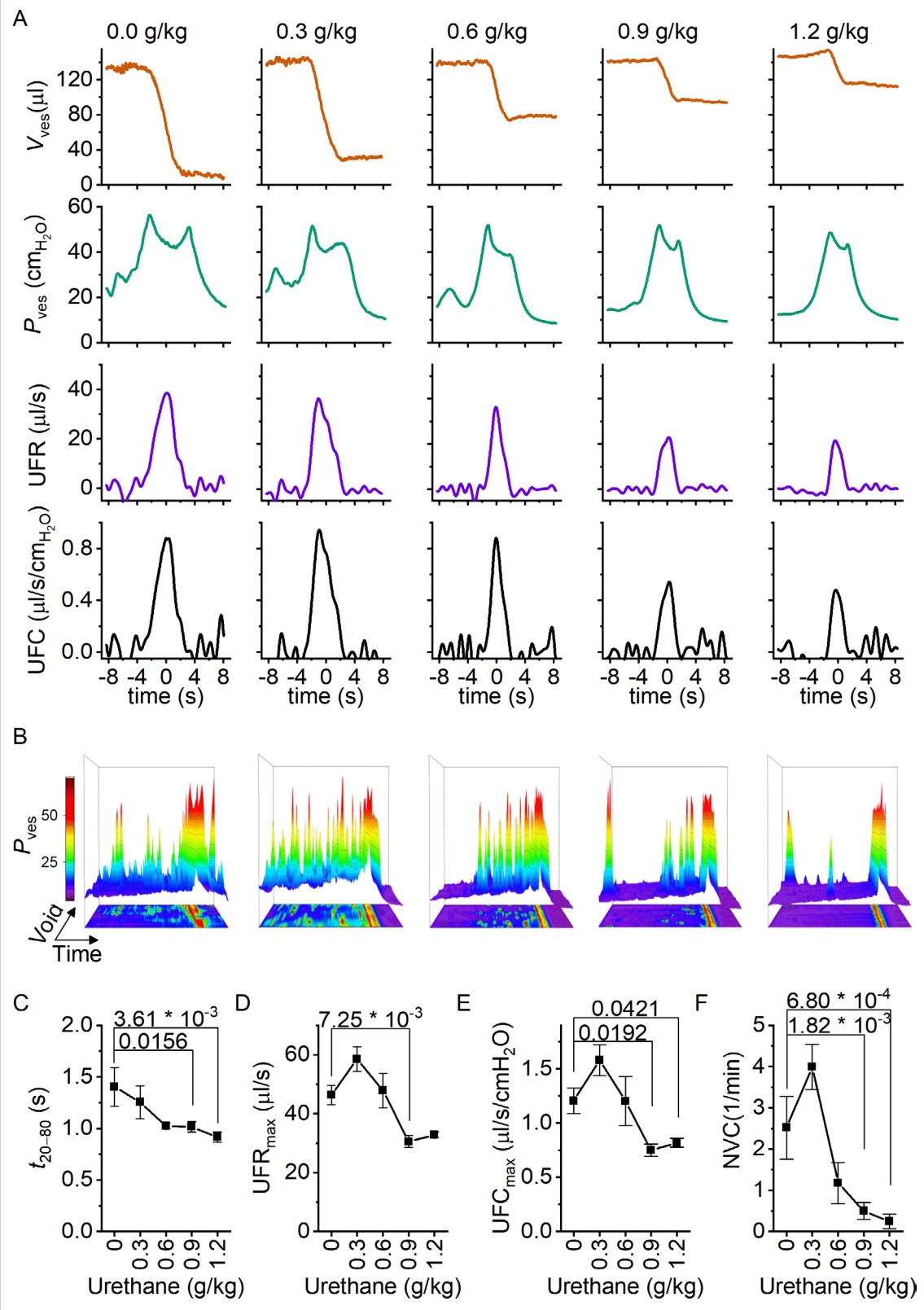

**Figure 3.** Dose-dependent effects of urethane on the voiding process. (**A**) Zoomed-in examples of average bladder volume, bladder pressure, urethral flow rate, and urethral flow conductance at increasing doses of urethane in catheterized mice. (**B**) Combined 2D/3D pressure plots showing 100 s of intravesical pressure changes aligned to a voiding contraction at time point 80 s from a single animal at the indicated dose. Each row represents the pressure during one individual void. The color scale on the right represents the intravesical pressure (in cmH$_2$O). (**C–F**) Assessment of the effect of

*Figure 3 continued on next page*

*Figure 3 continued*

urethane on void duration, urethral flow rate, urethral flow conductance, and nonvoiding contractions (mean ± standard error of the mean [SEM]). One-way repeated measures analysis of variance (ANOVA) was used to test for statistically significant differences with the 0.0 g/kg condition.

urethane dose up to a final 1.2 g/kg, which is a standard dose in LUT research. Urethane did not affect intravesical peak pressure during voiding, but dose-dependently suppressed nonvoiding contractions, in line with earlier studies (*Cheng et al., 1995*). In addition, urethane caused a strong and dose-dependent reduction of the voiding efficiency. Indeed, whereas in the awake animals, the bladder was completely emptied at every void, RV increased with mounting doses of urethane, causing voiding efficiency to decrease to around 50% at the highest dose. As these findings conform to previously published data on bladder volume and voiding efficiency in urethane-anesthetized mice, the suppressive effect of urethane on voiding cannot be attributed to gradual fatigue of the detrusor muscle in the course of the experiment (*Franken et al., 2021*). Detailed analysis of individual voids allowed us to attribute the reduction in voiding efficiency to both shorter void durations and a reduction in maximal UFC. These results indicate that urethane reduces the duration and extent of urethral relaxation during a void, leading to a pronounced reduction in voiding efficiency. Throughout these experiments, we observed that animals receiving the lowest dose of urethane (0.3 g/kg) were awake and behaving, but displayed a more tranquil behavior than nonanesthetized animals. Likewise, both pressure and volume data showed less artifacts in these animals treated with 0.3 g/kg, and there was a trend – albeit not statistically significant – for larger BC, longer ICI, and increased UFR and UFC when compared to nonanesthetized animals. Possibly, the analgesia provided by this low dose of urethane may reduce bladder pain caused by the surgical catheter implantation, thereby improving bladder function. Therefore, the use of urethane at a dose of 0.3 g/kg may be considered a good compromise between experimental/analytical ease and physiological conditions.

In order to measure bladder function in a noninvasive manner, we developed fluoroscopic volumetry, whereby subcutaneously administered contrast allowed fluoroscopic imaging of the bladder and monitoring of bladder volume during physiological filling, eliminating the need of catheter implantation. In line with the findings in the catheterized animals, we observed complete voiding in awake animals and a strong reduction of the voiding efficiency after urethane anesthesia. Notably, we also found a striking, fourfold larger BC when compared to mice undergoing videocystometry acutely after surgical implantation of the catheter, both in awake and anesthetized animals. Concurrent with the larger bladder capacities, we also measured a larger voided volume during individual voids in the noncatheterized animals, which we could attribute to longer void durations and larger peak UFRs. Surprisingly, whereas urethane caused a shortening of the void duration in catheterized animals, it lead to longer voids in the nonoperated animals – future research is warranted to elucidate the origin of this differential effect.

The above findings indicate that the use of a surgically implanted suprapubic catheter to infuse the mouse bladder has by itself a dramatic effect on bladder function. Several mechanisms may contribute to the fourfold reduced BC. First, the implantation of a catheter into the bladder dome and fixation using a purse-string suture inevitably leads to a reduction of the area of the bladder wall and thus the bladder volume. However, since the affected surface generally does not exceed 10% of the bladder wall, this surface reduction may only explain a small part (<20%) of the capacity reduction. Second, the surgical implantation will inevitably induce local bladder irritation and inflammation, which may contribute to the observed reduction in BC (*Boudes et al., 2011*). Whereas we performed our measurements within hours following surgery, earlier studies have used a 7-day recovery period in between surgery and cystometry to reduce the influence of inflammation on bladder function (*Yao et al., 2019*; *Mann-Gow et al., 2017*; *Cornelissen et al., 2008*). Notwithstanding the introduction of recovery period, these studies report voided volumes in the range of 50–140 µl for adult C57BL/6J mice (*Yao et al., 2019*; *Mann-Gow et al., 2017*; *Cornelissen et al., 2008*), similar to the values we obtained acutely after surgery, and substantially lower than the mean voided volume of 380 µl in the awake, noncatheterized animals. Lastly, in the fluoroscopic volumetry experiments on furosemide-treated mice we measured a filling rate of ~7 µl/min, which is lower than the 20 µl/min that we used in the videocystometry experiments, but higher than physiological rate of urine production in freely behaving mice of ~1–2 µl/min (*Meneton et al., 2000*). Further research is needed to establish the influence of the bladder filling rate on voiding behavior. These findings raise an important caveat when

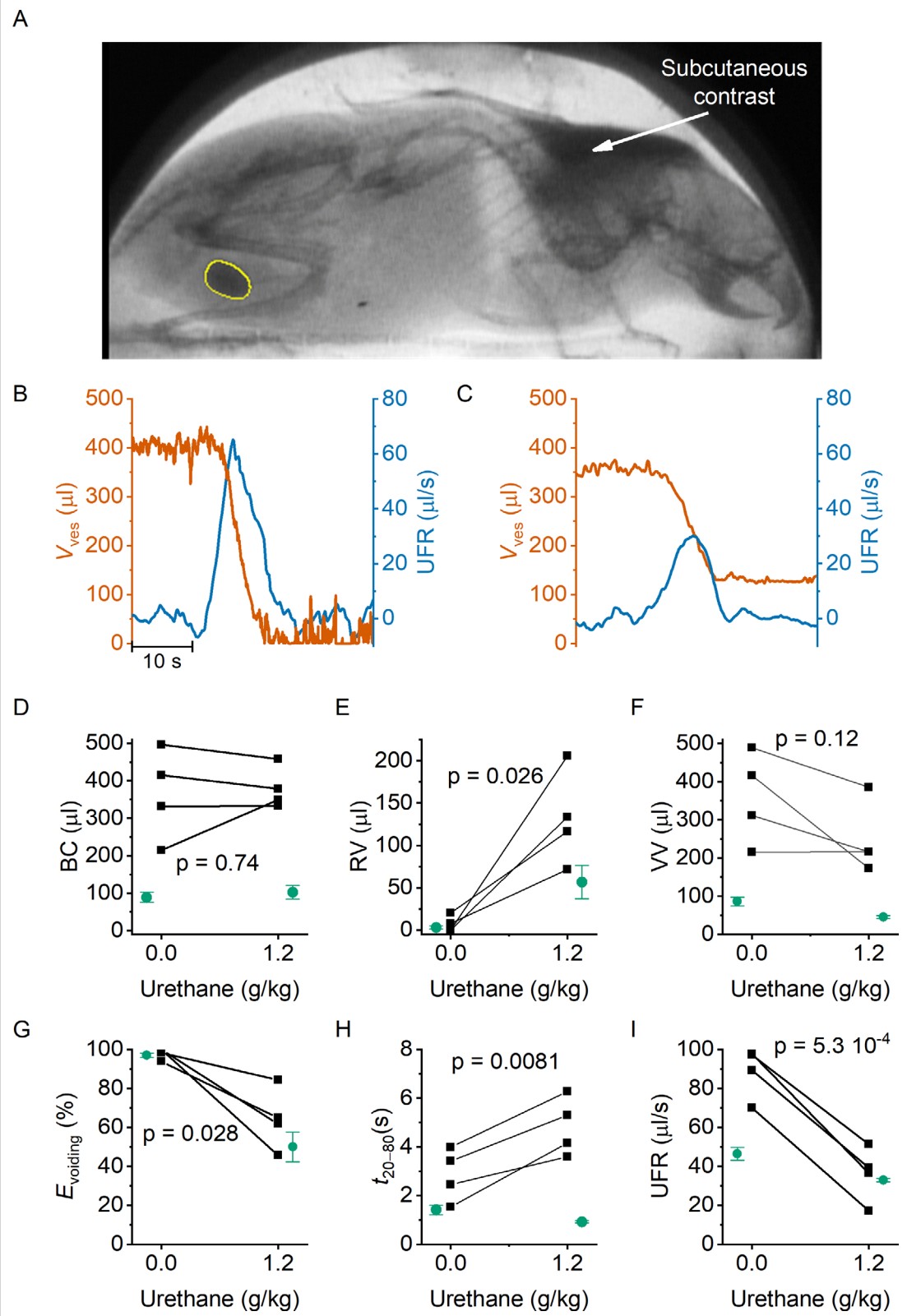

**Figure 4.** Noninvasive fluoroscopic volumetry. (**A**) Fluoroscopic image showing the presence of a contrast agent in the scruff (arrow) and in the bladder of an awake and nonrestrained mouse, as well as the automatic annotation of the bladder border. (**B–C**) Zoomed-in example of bladder volume and urethral flow rate in an animal that did not receive any urethane anesthesia and in the same animal after dosing at 1.2 g/kg. Note that, due to the complete bladder emptying and slow filling in the absence of urethane, the void is followed by a period of low-accuracy volume measurement. (**D–I**)

*Figure 4 continued on next page*

*Figure 4 continued*

Comparison of the indicated voiding parameters in animals without an implanted catheter (black), before and after urethane dosing. In green, we show the corresponding values from catheterized animals reproduced from *Figures 2 and 3*. Paired *t*-tests comparing the values in the noncatheterized animals before and after urethane dosing yielded the indicated p values.

interpreting (video)cystometric results and highlight the usefulness of fluoroscopic volumetry to study bladder function noninvasively, albeit at the expense of intravesical pressure recordings.

In conclusion, we have combined fluoroscopy with a machine learning-assisted analysis method to monitor the function of the LUT at high temporal resolution in mice, thereby providing a powerful approach for the noninvasive study of bladder function in awake, behaving mice.

# Materials and methods

## Key resources table

| Reagent type (species) or resource | Designation | Source or reference | Identifiers | Additional information |
|---|---|---|---|---|
| Strain, strain background (*Mus musculus*) | Wild-type mice | Janvier | C57BL/6J | 12- to 16-week-old females |
| Chemical compound, drug | Isoflurane | Dechra veterinary products | Iso-vet | Anesthesia |
| Chemical compound, drug | Carprofen | Zoetis | Rimadyl | Analgesia |
| Chemical compound, drug | Surgical skin glue | Vygon | | |
| Chemical compound, drug | Iomeprol | Bracco Imaging Europe | Iomeron 250 | Contrast agent |
| Chemical compound, drug | Iodixanol | GE Healthcare | Visipaque 320 mg I/ml | Contrast agent |
| Chemical compound, drug | Urethane | Sigma-Aldrich | | Anesthesia |
| Chemical compound, drug | Furosemide | Sanofi | Lasix | Diuretic |
| Software, algorithm | ImageJ/FIJI | NIH | RRID:SCR_002865 | Image processing |
| Software, algorithm | Igor Pro | Wavemetrics | RRID:SCR_000325 | Data processing |
| Software, algorithm | Origin 9.0 | Originlab | RRID:SCR_014212 | Data processing |
| Software, algorithm | NIS-Elements; NIS.ai | Nikon | RRID:SCR_014329 | Image processing |

## Animals

Experiments were conducted on 12- to 16-week-old female C57BL/6J mice (Janvier). Mice were housed in filter-top cages in a conventional facility at 21°C on a 12 hr light–dark cycle with unrestricted access to food and water. After each experiment, all animals were sacrificed due to the toxicity associated with urethane anesthesia. All animal experiments were carried out after approval of the Ethical Committee Laboratory Animals of the Faculty of Biomedical Sciences of the KU Leuven under project number P035/2018.

## Surgery and experimental setup

### Urethane dose escalation

Videocystometry and suprapubic catheter implantation were performed as described earlier (*Franken et al., 2021*; *Uvin et al., 2012*). In short, a PE-50 catheter was implanted into the bladder dome of five female mice through an abdominal midline incision while under 2% isoflurane anesthesia (Iso-vet, Dechra veterinary products, Bladel, The Netherlands). At the start of surgery, the anesthetized animals received carprofen (5 mg/kg bodyweight; Zoetis, Louvain-la-Neuve, Belgium) diluted in NaCl 0.9% by SC injection to achieve analgesia during videocystometry. At the end of surgery, the implanted catheter was tunneled subcutaneously and exteriorized at the scruff. At the point of exteriorization, the catheter was secured to the skin using surgical skin glue (Derma+flex, Vygon, Brussels, Belgium), to prevent dislocation. After the animals recovered from surgery, they were placed in a small radiolucent container (4 cm × 3 cm × 7 cm) in which the animals were not restrained. Next, the container was placed in an X-ray-based small animal fluoroscopy system (LabScopeTM, Glenbrook Technologies, New Jersey, USA), and the PE-50 catheter was connected to a pressure sensor and to an infusion pump through a three-way stopcock. Videocystometry was performed with an

infusion of an iodine-based intravesical contrast solution (50% Iomeron 250, Bracco Imaging Europe, Wavre, Belgium) at a rate of 20 µl/s. Simultaneously, pressure measurements were recorded at 50 Hz, and bladder images were acquired at a frame rate of 30 frames per second. Videocystometry was performed for 40 min, including a habituation period of 10 min. Next, a dose of 0.3 g/kg bodyweight urethane (Sigma-Aldrich, Diegem, Belgium) was injected subcutaneously. Ten minutes after the injection, videocystometry was performed for another 40 min. This procedure was repeated until a cumulative urethane dose of 1.2 g/kg was reached; at this dose, videocystometry was performed one last time, after which the experiment was terminated.

## Catheter-free bladder imaging

Thirty minutes prior to fluoroscopy, mice were injected subcutaneously with the diuretic furosemide (15 mg/kg, 1 mg/ml solution, Sanofi, Paris, France), to increase the diuresis and thereby the filling rate of the bladder (*Monjotin et al., 2017*), and with an iodine-containing nonionic radiocontrast agent iodixanol (12 mg I/kg bodyweight, Visipaque 320 mg I/ml; GE Healthcare, Chicago, IL, USA). The contrast agent was injected in the scruff area, to prevent superposition of the injected contrast solution with the bladder. Time-lapse fluoroscopy was performed on awake, freely moving mice, without bladder catheter in situ, during multiple cycles of 40 min. Next, urethane was injected subcutaneously at a dose of 1.2 g/kg bodyweight, after which the bladder was again imaged until a void occurred.

## **Data processing and analysis**

As the animals were awake and freely moving, background subtraction was not possible, and the bladder volume could not be determined based on image intensity as in previous experiments (*Franken et al., 2021*). Therefore, we identified the bladder and traced the bladder circumference in each image, and calculated bladder volume based on the long and short radii of the identified bladder area, assuming the bladder approaches a prolate spheroid, as established in our earlier work (*Franken et al., 2021*):

$$\text{Vves} = \tfrac{4}{3}\pi \times \text{long radius} \times \text{short radius}^2.$$

Images were saved as Audio Video Interleave file (AVI), loaded into FIJI (ImageJ) (*Schindelin et al., 2012*), and saved as an image sequence. This image sequence was loaded into NIS Elements-AR 5.30.01 (Nikon, Tokyo, Japan) and smoothed using a rolling average, to enhance image quality for bladder delineation. Based on simulations using artificial images of an idealized spheroid bladder, we determined that a rolling average of 15 frames provided an overall suitable image quality with limited influence on key parameters such as the timing and amplitude of the UFR (see *Figure 1—figure supplement 1*). To identify the bladder border in large sets of time-lapse fluoroscopy images, we used Segment.ai, an AI module that is a part of the NIS-Elements NIS.ai suite (Nikon, Tokyo, Japan), a commercially available (licensed) software. We trained this neural network using a set of manually annotated fluoroscopy images in which the contrast-filled bladder was imaged from various angles and at different stages of the filling/voiding cycle. The neural network was then applied to identify the bladder in similar but different datasets. It was repeatedly retrained using images where the bladder was not properly identified until satisfactory identification of the bladder border was achieved in all experiments (*Figure 1A*). Additional filtering steps were applied to detect only one closed object (the biggest) as the bladder. The NIS GA3 script used for this purpose together with a description can be found here (https://doi.org/10.6084/m9.figshare.c.6004234). The neural network was then used to generate an image sequence with an overlay tracing the bladder border, used for quantifying the long and short axes of the bladder and the bladder area, for large sets of time-lapse fluoroscopy images.

To benchmark the performance of the neural network, 20% of the ground truth annotations were kept aside, as described in the BIAFLOWS collaborative framework for quality control of object segmentation (*Rubens et al., 2020*), and used to calculate the DISI:

$$\text{DISI} = \frac{2 \times (X \cap Y)}{X + Y},$$

where $X$ is the area of the ground truth binary mask, $Y$ the area of the prediction binary mask, and $X \cap Y$ the area of overlap between the ground truth and prediction masks (*Dice, 1945*).

To comply with recent publications on AI reporting (*Walsh et al., 2021*), the trained neural network, the user manual, the training dataset, and the annotations can be found here (https://doi.org/10.6084/m9.figshare.c.6004234). As the network will only be usable from within NIS, with our training data and the annotations (.nd2 file) researchers can apply a similar strategy to detect the bladder in awake mice using a comparable imaging setup providing similar quality images. The user manual describes in detail the requirements and workflow on how the different scripts can be opened and implemented in NIS-Elements.

From the time course of $V_{ves}$, we determined a set of parameters related to volume alterations during urine storage and voiding (BC, RV, and $E_{voiding}$) and to urine flow during voiding (UFR and UFC), as established earlier (*Franken et al., 2021*), with an additional Savitzky–Golay digital filter applied to $V_{ves}$ for data smoothing before differentiation to calculate UFR. As a measure of void duration, we determined $t_{20-80}$, which represents the interval between the time points at which the bladder volume was reduced by 20% and 80% of the total void volume.

Raw data for *Figures 1–4* are available via https://doi.org/10.6084/m9.figshare.19826050.v1.

## Statistics

Statistical analysis and graphing were performed in OriginPro 9.0 (Originlab Corporation, Northampton, MA, USA). The Shapiro–Wilk test was used to assess normality of the data. Paired Student's *t*-test, Student's *t*-test, and one-way repeated measures analysis of variance were used for analysis of the data. Except where indicated otherwise, all summary data are reported as mean ± standard error. The sample sizes were calculated based on our earlier work, which yielded that a sample size of four to five animals was sufficient to detect a 40% change in key parameters such as $E_{voiding}$ and $UFR_{max}$ with a power of 80% and a significance level of p < 0.05. Animals were randomly assigned to each experimental condition. All measurements were performed on distinct animals; no animals were used in multiple experiments. Analysis was performed automatically, eliminating the need for blinding.

## Acknowledgements

We thank Benjamin Pavie for help with the statistical benchmarking of the machine-learning protocol, Andrei Segal for assistance with data management and processing, and all members of the Laboratory of Ion Channel Research for helpful comments and discussion.

## Additional information

### Funding

| Funder | Grant reference number | Author |
| --- | --- | --- |
| Fonds Wetenschappelijk Onderzoek | I001322N | Sebastian Munck |
| Fonds Wetenschappelijk Onderzoek | I000321N | Sebastian Munck |
| KU Leuven | KA/20/085 | Sebastian Munck |
| KU Leuven | IDN/19/039 | Sebastian Munck |
| Fonds Wetenschappelijk Onderzoek | Senior Clinical Investigator fellowship | Wouter Everaerts |
| Fonds Wetenschappelijk Onderzoek | G066322N | Wouter Everaerts |
| KU Leuven | C24M/21/028 | Wouter Everaerts |
| Queen Elisabeth Medical Foundation | | Thomas Voets |
| Vlaams Instituut voor Biotechnologie | Unrestricted grant | Thomas Voets |

| Funder | Grant reference number | Author |
|---|---|---|
| Fonds Wetenschappelijk Onderzoek | G0B7620N | Thomas Voets |

The funders had no role in study design, data collection, and interpretation, or the decision to submit the work for publication.

## Author contributions

Helene De Bruyn, Conceptualization, Data curation, Formal analysis, Investigation, Visualization, Writing – original draft; Nikky Corthout, Software, Formal analysis, Writing – review and editing; Sebastian Munck, Software, Methodology, Writing – review and editing; Wouter Everaerts, Conceptualization, Funding acquisition, Writing – review and editing; Thomas Voets, Conceptualization, Software, Formal analysis, Supervision, Funding acquisition, Writing – original draft

## Author ORCIDs

Helene De Bruyn http://orcid.org/0000-0002-7072-4181
Nikky Corthout http://orcid.org/0000-0003-1176-7277
Sebastian Munck http://orcid.org/0000-0002-5182-5358
Wouter Everaerts http://orcid.org/0000-0002-3157-7115
Thomas Voets http://orcid.org/0000-0001-5526-5821

## Ethics

All animal experiments were carried out after approval of the Ethical Committee Laboratory Animals of the Faculty of Biomedical Sciences of the KU Leuven under project number P035/2018.

## Decision letter and Author response

Decision letter https://doi.org/10.7554/eLife.79378.sa1
Author response https://doi.org/10.7554/eLife.79378.sa2

# Additional files

## Supplementary files

• MDAR checklist

## Data availability

Raw data for Figures 1-4 are available via https://doi.org/10.6084/m9.figshare.19826050.v1.

The following dataset was generated:

| Author(s) | Year | Dataset title | Dataset URL | Database and Identifier |
|---|---|---|---|---|
| Voets T | 2022 | Raw data for Figures | https://doi.org/10.6084/m9.figshare.19826050.v1 | figshare, 10.6084/m9.figshare.19826050.v1 |

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
