## [Editor Report]

The described technique allows for accurate and detailed measurement of intravesical pressures, urethral flow rate, inter-contraction intervals, and residual bladder volume without the need for catheter implantation, and in anesthetized and awake mice. This method has the potential for broad application ranging from the elucidation of molecular mechanisms of bladder control to drug development for LUT dysfunction. The reporting on the AI training process, a user manual and the trained neural network makes this machine-learning approach to image-analysis a useful tool for LUT studies in other labs.

---

## [Decision Letter]

**Decision letter after peer review:**

Thank you for submitting your article "Machine learning-assisted fluoroscopy of bladder function in awake mice" for consideration by *eLife*. Your article has been reviewed by 2 peer reviewers, and the evaluation has been overseen by a Reviewing Editor and Martin Pollak as the Senior Editor. The following individual involved in review of your submission has agreed to reveal their identity: Michael Ruggieri (Reviewer #1).

Upon addressing the essential revisions and the reviewer's concerns we would recommend this manuscript to be a "Short Report" type of article for publication in *eLife*.

Essential revisions:

1) An important finding in this manuscript is that the bladder capacity is drastically reduced in catheterized animals, both in awake and anaesthetized conditions. It is unclear however if this ~4 fold reduction is (mostly) due to acute injury/ inflammation as a result of the bladder catheter implantation surgery or of catheterization in general.

Furthermore, cystometry in awake animals often involves animals having the catheter tunneled subcutaneously, exteriorized (into a harness for easy-access) to be connected to an infusion system. Most researchers performing awake cystometry allow up to 7 days before starting the recordings.

– To support your claim that bladder capacity is affected by catheterization rather than due to catheter implantation surgery: can you either (i) evaluate the bladder capacity, with catheter implanted, at a time that the animal is recovered from the surgery and any injury/ inflammation has subsided (see reviewer #2 comments)? Or (ii) incorporate the word 'acutely' in this sentence? "Bladder capacity was decreased ~4-fold when cystometry was performed acutely after surgery, in awake and anaesthetized mice".

Also, we would like to request additional details to the description of how the PE50 tubing implanted into the bladder exited the animal for allowing freedom of movement (in the awake un-restrained catheterized condition).

2) The ai training process and the use of the neural network would benefit from more description:

– "The neural network was repeatedly retrained until satisfactory identification of the bladder was achieved": Please provide some statistics about the accuracy and failure rate of bladder identification.

– Can the software (as shared here) be readily used in NIS Elements or other commercially available environments, or does the machine learning algorithm have to be set up freshly in any lab through the use of the Nikon ai Suite?

– Could you address and incorporate into the manuscript the machine learning-related questions of the reviewers?

3) Please revise text, Figures and Figure legends in line with reviewer's recommendations.

*Reviewer #1 (Recommendations for the authors):*

Page 5 bottom – second paragraph of results: Suggest restating the criteria used to define a successful identification of the bladder border such as: "The machine-learning protocol was able to successfully identify the bladder border compared to manual bladder wall delineation in timelapse image sequences, irrespective…"

Throughout the manuscript the term "catheterization" is used which many may confuse with urethral catheterization and this was not done to these mice. Suggest replacing all or most instances of "catheterization" with "suprapubic catheterization".

Page 4 middle of the first paragraph of discussion: change "chirurgical" to "surgical".

Page 8 middle: change "IC" to "ICI".

Page 9, line 1: Suggest changing "We performed videocystometry in mice," with "We performed videocystometry in suprapubic catheter implanted mice,".

Page 10 middle of last paragraph of discussion: The authors could easily eliminate one of these possibilities by performing the fluoroscopic volumetry on suprapubic catheter implanted animals that are filled at approximately the same rate as they found to be the furosemide induced filling rate in the non-catheterized mice.

Page 10 "Surgery and experimental setup:": Suggest adding additional details to this description of the how the PE50 tubing implanted into the bladder exited the animal while allowing freedom of movement (use of a tether?).

Figure 1AandC: suggest increasing the resolution of these images to 600 DPI or higher. Figure 1BandC: suggest using a higher contrast color for the identified bladder border such as white or bright yellow as opposed to the blue color used in the figure.

Figure 2C-F: Suggest stating what the blue horizontal line indicate or removing these lines.

Figure 2 legend: suggest changing the first line to "Short excerpts of simultaneous bladder volume and intravesical pressure recordings traces at increasing cumulative doses of urethane in suprapubic catheter implanted animals."

Figure 3A: Because urethral flow conductance is defined by the authors as urethral flow divided by intravesical pressure, suggest including an additional row of graphs between the PVES and UFC rows for urethral flow rate alone (UFR (ml/s)) so the readers may better understand this novel measure of urethral flow mechanics.

Figure 3B: The description of these "2D/3D" graphs is confusing. Suggest including labels for each of the axes. If only 100 seconds of time is displayed on the horizontal axes of each of these graphs, and pressure is on the vertical axes then it seems that neither the color scale nor the third axes of the 3D plots are necessary and only cause confusion. The legend for figure 3B is also confusing. The statement "Each row represents the pressure during one individual void." is unclear because there is only one row of 2D/3D graphs in figure 3B. Suggest changing the legend to something like "Representative pressure plots from an individual animal showing 100 seconds of pressure recordings aligned to a voiding contraction at 80 seconds."

Figure 3D: It seems that this is the maximal urethral flow rate and if so, the vertical axis label needs to be changed to "UFR max (ml/s)".

*Reviewer #2 (Recommendations for the authors):*

It is an excellent approach to not instrument the mice and yet get the functional data of the bladder dynamics. It is a breakthrough in the field. The dramatically larger BC in non-catheterized mice bladder compared to catheterized bladder can be a cause for concern for many researchers who perform cystometry. It is of particular interest to note that the BC of awake catheterized mice is same as that of a full dose urethane supplemented, catheterized mice. The authors have given various hypothesis as to why this would happen. One of the hypotheses relates to supraphysiological-filling rate which might reduce bladder capacity as was shown previously (ref, 10); However, BC was not measured in reference 10, sentence needs to be corrected.

Since in vivo cystometry in awake rodents (implanted bladder catheter) is widely practiced, it begs the important question, as to why does catheterization affect the functional bladder parameters. Is bladder injury the main cause of the decreases in bladder metrics? The authors have mentioned, bladder inflammation may be a primary factor in decreasing the BC. Since one of the important conclusions of the manuscript is the ~ 4 fold reduction of BC in mice that were catheterized, compared to non-catheterized, I believe it is important to address this issue, as it would give clarity on use of catheter for researchers in the field of urology.

To address if it is the bladder injury/ inflammation or the mere presence of a catheter that is the cause of reduced BC, the authors could perform a sham surgery where a catheter is inserted into bladder as was done in the catheterized mice, but catheter is immediately removed after insertion into bladder and puncture is sutured. This would give insight into effect of bladder injury on BC.

To check if the presence of the catheter is cause of reduced BC, the catheter could be implanted in the bladder as was done previously, let the animal recover for about 6-7 days and then perform videocystometry. The catheter can be tunneled subcutaneously and exteriorized around the neck area. Any bladder injury or inflammation should be healed by that time. Another option is to implant the catheter, seal the open end of the catheter outside the bladder and leave it in the subcutaneous space. Then after a 6-7 day recovery perform the same experiments as was done for mice without catheter.

It was shown that bladder inflammation subsides within a week after implantation of the catheter (refs URL: https://www.jove.com/video/55588 and Yao et al., Front Neuroscience. 2019 Jun 25;13:663).

Urethane and reduced voiding efficiency: Authors attribute reduced voiding efficient to shorted voiding duration and reduction in maximal UFC. Since UFC is urine flow rate/ unit pressure, it would suggest that UFC decreased, either due to an increase in resistance along the urine flow path or that the muscle contraction is weak. Further, it's unclear how shortened voiding duration per se would affect voiding efficiency since voiding efficiency does not involve time. Perhaps the authors meant to say incomplete or partial voiding leads to a short void duration and therefore decreased voiding efficiency.

In figure 3., although void duration decreases as urethane dose is increased, UFR and UFC increases at 0.3 g/kg urethane. Would authors like to comment on it?

How many times did the non-catheterized mice void during the 40 minutes of recording when the contrast agent was delivered in the neck region?

Questions about AI:

1. "AI was trained until satisfactory identification of the bladder was achieved" – can the authors give some statistics about the accuracy and failure rate of bladder identification.

2. After a void when the bladder is almost empty AI cannot recognize the bladder border. Therefore, from what point of bladder filling does AI start to reliably draw the bladder contours? Similarly, during a void, till what percentage of bladder volume does it reliably draw the bladder outlines?

3. With a camera frame rate of 30, FPS and a void duration of 1-2 seconds, would smoothing using rolling average of 15 frames, lead to loss to temporal resolution or a shift in time of peak pressure?

4. The manuscript mentions "we provide the trained neural network for other researchers, including a description on how to use it in within the NIS environment (https://doi.org/10.6084/m9.figshare.c.6004234). We also provide the training data and the annotations, such that researchers can apply a similar strategy to detect the bladder in awake mice using a comparable imaging setup providing similar quality images (https://doi.org/10.6084/m9.figshare.c.6004234)."

The figshare site shows some files containing data and software. Is the software independent or it has a dependency on having other programs from Nikon? Information on how to use or run it is unclear.

Figure 1. In the figure legend please indicate if these (B and C) are the awake or urethane anaesthetized mice. Panel B picture 3 of bladder shows significant residual volume, would that be urethane anaesthetized?

Figure 3. In figure legend, indicate that the mice are catheterized.

Figure 4. The legend mentions that urethane catheterized mice values taken from figure 2 and 3. However, the residual volume (RV) of urethane dosed mice (catheter, green dot) in figure 4E is < 25 ul, whereas in figure 2D urethane dosed (catheter, blue) mice RV is ~ 60 ul.

---

## [Author Response]

Essential revisions:1) An important finding in this manuscript is that the bladder capacity is drastically reduced in catheterized animals, both in awake and anaesthetized conditions. It is unclear however if this ~4 fold reduction is (mostly) due to acute injury/ inflammation as a result of the bladder catheter implantation surgery or of catheterization in general.Furthermore, cystometry in awake animals often involves animals having the catheter tunneled subcutaneously, exteriorized (into a harness for easy-access) to be connected to an infusion system. Most researchers performing awake cystometry allow up to 7 days before starting the recordings.– To support your claim that bladder capacity is affected by catheterization rather than due to catheter implantation surgery: can you either (i) evaluate the bladder capacity, with catheter implanted, at a time that the animal is recovered from the surgery and any injury/ inflammation has subsided (see reviewer #2 comments)? Or (ii) incorporate the word 'acutely' in this sentence? "Bladder capacity was decreased ~4-fold when cystometry was performed acutely after surgery, in awake and anaesthetized mice".Also, we would like to request additional details to the description of how the PE50 tubing implanted into the bladder exited the animal for allowing freedom of movement (in the awake un-restrained catheterized condition).

The point regarding catheterization versus injury was well taken, and we have addressed this issue further at several instances in the revised manuscript. As outlined in our response to reviewer #2, we are not in the position to perform experiments using animals with a chronically implanted catheter within a reasonable period, since we currently don’t have ethical approval for such chronic experiments. We now clearly indicate in the abstract and discussion that our experiments were done “acutely” after the surgery. Moreover, we provide a more in-depth discussion of the potential effects of acute versus chronic catheter implantation in the Discussion section. Finally, in the methods section, we provide details regarding the implantation and tunneling of the catheter in the freely moving animals.

2) The ai training process and the use of the neural network would benefit from more description:– "The neural network was repeatedly retrained until satisfactory identification of the bladder was achieved": Please provide some statistics about the accuracy and failure rate of bladder identification.– Can the software (as shared here) be readily used in NIS Elements or other commercially available environments, or does the machine learning algorithm have to be set up freshly in any lab through the use of the Nikon ai Suite?– Could you address and incorporate into the manuscript the machine learning-related questions of the reviewers?

We have now included statistics on the accuracy and detection threshold of neural network protocol, and clarified how the procedures can be implemented in other labs.

3) Please revise text, Figures and Figure legends in line with reviewer's recommendations.

We have addressed all the reviewer’s comments and suggestions, as outlined below.

Reviewer #1 (Recommendations for the authors):Page 5 bottom – second paragraph of results: Suggest restating the criteria used to define a successful identification of the bladder border such as: "The machine-learning protocol was able to successfully identify the bladder border compared to manual bladder wall delineation in timelapse image sequences, irrespective…"

This has been changed according to the suggestion.

Throughout the manuscript the term "catheterization" is used which many may confuse with urethral catheterization and this was not done to these mice. Suggest replacing all or most instances of "catheterization" with "suprapubic catheterization".

We have now indicated in several instances that we used a suprapubic catheter.

Page 4 middle of the first paragraph of discussion: change "chirurgical" to "surgical".

This has been changed according to the suggestion.

Page 8 middle: change "IC" to "ICI".

This has been changed according to the suggestion.

Page 9, line 1: Suggest changing "We performed videocystometry in mice," with "We performed videocystometry in suprapubic catheter implanted mice,".

This has been changed according to the suggestion.

Page 10 middle of last paragraph of discussion: The authors could easily eliminate one of these possibilities by performing the fluoroscopic volumetry on suprapubic catheter implanted animals that are filled at approximately the same rate as they found to be the furosemide induced filling rate in the non-catheterized mice.

We now include in the Results section a quantification of the filling rate during fluoroscopic volumetry, which yielded a mean value of ~7 μl/min. There is currently no good literature on the influence of filling rate on bladder capacity and voiding in mice. In the future, we plan long-term fluoroscopy on mice with normal urine production, but these experiments fall beyond the scope and time-frame of this study.

Page 10 "Surgery and experimental setup:": Suggest adding additional details to this description of the how the PE50 tubing implanted into the bladder exited the animal while allowing freedom of movement (use of a tether?).

Additional details have been added.

Figure 1AandC: suggest increasing the resolution of these images to 600 DPI or higher.

We now provide high-resolution images.

Figure 1BandC: suggest using a higher contrast color for the identified bladder border such as white or bright yellow as opposed to the blue color used in the figure.

This has been improved, using bright yellow. We also include two videos to illustrate the border detection.

Figure 2C-F: Suggest stating what the blue horizontal line indicate or removing these lines.

The blue line indicates the dose of urethane received by the animals corresponding to the blue symbols. This is now clarified in the legend.

Figure 2 legend: suggest changing the first line to "Short excerpts of simultaneous bladder volume and intravesical pressure recordings traces at increasing cumulative doses of urethane in suprapubic catheter implanted animals."

This has been changed according to the suggestion.

Figure 3A: Because urethral flow conductance is defined by the authors as urethral flow divided by intravesical pressure, suggest including an additional row of graphs between the PVES and UFC rows for urethral flow rate alone (UFR (ml/s)) so the readers may better understand this novel measure of urethral flow mechanics.

A row with the corresponding UFR traces has been included in Figure 3. The previous version also showed idealized traces for UFC. These have been replaced by the raw data.

Figure 3B: The description of these "2D/3D" graphs is confusing. Suggest including labels for each of the axes. If only 100 seconds of time is displayed on the horizontal axes of each of these graphs, and pressure is on the vertical axes then it seems that neither the color scale nor the third axes of the 3D plots are necessary and only cause confusion. The legend for figure 3B is also confusing. The statement "Each row represents the pressure during one individual void." is unclear because there is only one row of 2D/3D graphs in figure 3B. Suggest changing the legend to something like "Representative pressure plots from an individual animal showing 100 seconds of pressure recordings aligned to a voiding contraction at 80 seconds."

We provided further clarification of these plots in the legend.

Figure 3D: It seems that this is the maximal urethral flow rate and if so, the vertical axis label needs to be changed to "UFR max (ml/s)".

This has been corrected in the revised manuscript.

Reviewer #2 (Recommendations for the authors):It is an excellent approach to not instrument the mice and yet get the functional data of the bladder dynamics.

We thank the reviewer for the positive evaluation and very helpful comments.

It is a breakthrough in the field. The dramatically larger BC in non-catheterized mice bladder compared to catheterized bladder can be a cause for concern for many researchers who perform cystometry. It is of particular interest to note that the BC of awake catheterized mice is same as that of a full dose urethane supplemented, catheterized mice. The authors have given various hypothesis as to why this would happen. One of the hypotheses relates to supraphysiological-filling rate which might reduce bladder capacity as was shown previously (ref, 10); However, BC was not measured in reference 10, sentence needs to be corrected.

We have removed this reference from the paper. On scrutiny, this paper does not provide helpful information on the effect of infusion speed on bladder capacity, and some of the calculations in the paper are questionable. We tried to back-calculate the influence of infusion speed on bladder capacity from the provided infusion speeds and ICI data in this paper, and actually found that voided volumes increase with increasing infusion speed, suggesting that also BC increases. However, since BC was indeed not directly calculated, we decided to remove the citation. In the discussion, we now leave it open whether the filling rate is a contributing factor.

Since in vivo cystometry in awake rodents (implanted bladder catheter) is widely practiced, it begs the important question, as to why does catheterization affect the functional bladder parameters. Is bladder injury the main cause of the decreases in bladder metrics? The authors have mentioned, bladder inflammation may be a primary factor in decreasing the BC. Since one of the important conclusions of the manuscript is the ~ 4 fold reduction of BC in mice that were catheterized, compared to non-catheterized, I believe it is important to address this issue, as it would give clarity on use of catheter for researchers in the field of urology.To address if it is the bladder injury/ inflammation or the mere presence of a catheter that is the cause of reduced BC, the authors could perform a sham surgery where a catheter is inserted into bladder as was done in the catheterized mice, but catheter is immediately removed after insertion into bladder and puncture is sutured. This would give insight into effect of bladder injury on BC.To check if the presence of the catheter is cause of reduced BC, the catheter could be implanted in the bladder as was done previously, let the animal recover for about 6-7 days and then perform videocystometry. The catheter can be tunneled subcutaneously and exteriorized around the neck area. Any bladder injury or inflammation should be healed by that time. Another option is to implant the catheter, seal the open end of the catheter outside the bladder and leave it in the subcutaneous space. Then after a 6-7 day recovery perform the same experiments as was done for mice without catheter.It was shown that bladder inflammation subsides within a week after implantation of the catheter (refs URL: https://www.jove.com/video/55588 and Yao et al., Front Neuroscience. 2019 Jun 25;13:663).

We agree with the reviewer that an experiment including a 7-day recovery period would be informative to estimate the contribution of surgery-induced inflammation to the reduction in bladder volume. However, we currently do not have ethical approval for such a chronic experiment, and obtaining such approval + performing the experiments would delay publication of our study by at least 6 months. However, we have included more discussion on the voided volumes that were found in studies performing cystometry in mice after a 7-day recovery period. These values are ranging between 50 and 140 μl, very similar to the values we report here for the acute measurements. We now also clarify that in our experiments the catheter was also tunneled and exteriorized at the neck area, as mentioned by the reviewer.

Urethane and reduced voiding efficiency: Authors attribute reduced voiding efficient to shorted voiding duration and reduction in maximal UFC. Since UFC is urine flow rate/ unit pressure, it would suggest that UFC decreased, either due to an increase in resistance along the urine flow path or that the muscle contraction is weak. Further, it's unclear how shortened voiding duration per se would affect voiding efficiency since voiding efficiency does not involve time. Perhaps the authors meant to say incomplete or partial voiding leads to a short void duration and therefore decreased voiding efficiency.

As described in our previous work (Franken et al., *Sci Adv.* 2021), and as generally used in clinical practice, we define voiding efficiency as the fraction of the bladder content that is expelled during a void. Reduced voiding efficiency can effectively be the result of a shortened void duration: if voiding stops before the bladder is emptied, the voiding efficiency will be reduced. Oppositely, if the void duration is unaltered but the urethral conductance/flow is reduced, there will be less urine expelled during the void, again leading to a reduced voiding efficiency. Here we see both, indicating that urethane affects the extent and duration of urethral relaxation. We have further clarified this in the manuscript.

In figure 3., although void duration decreases as urethane dose is increased, UFR and UFC increases at 0.3 g/kg urethane. Would authors like to comment on it?

This was already briefly addressed in the discussion. In the revised version, we added a sentence where we suggest that this may be due to the mild analgesia caused by low dose urethane.

How many times did the non-catheterized mice void during the 40 minutes of recording when the contrast agent was delivered in the neck region?

These data are now included in the Results section.

Questions about AI:1. "AI was trained until satisfactory identification of the bladder was achieved" – can the authors give some statistics about the accuracy and failure rate of bladder identification.

We benchmarked the accuracy of the bladder identification using the Dice similarity index (DISI), as described in the BIAFLOWS collaborative framework. This yielded a median value of 0.95, indicating excellent agreement between manual annotation and AI bladder identification. This is now described in the results and methods sections.

2. After a void when the bladder is almost empty AI cannot recognize the bladder border. Therefore, from what point of bladder filling does AI start to reliably draw the bladder contours? Similarly, during a void, till what percentage of bladder volume does it reliably draw the bladder outlines?

We included in the results statistics on the minimal bladder volume that could still be reliably detected, which was in the range of 4 μl.

3. With a camera frame rate of 30, FPS and a void duration of 1-2 seconds, would smoothing using rolling average of 15 frames, lead to loss to temporal resolution or a shift in time of peak pressure?

As we now explain in the manuscript, the rolling average of 15 images is a compromise between image quality and temporal resolution. We performed simulations with different numbers of frames for averaging, showing that at 15 frames the effect on the fastest voiding events is minimal, whereas using larger number of frames reduced the peak UFR and shifts time of the voiding peak (Figure 1 —figure supplement 1).

4. The manuscript mentions "we provide the trained neural network for other researchers, including a description on how to use it in within the NIS environment (https://doi.org/10.6084/m9.figshare.c.6004234). We also provide the training data and the annotations, such that researchers can apply a similar strategy to detect the bladder in awake mice using a comparable imaging setup providing similar quality images (https://doi.org/10.6084/m9.figshare.c.6004234)."The figshare site shows some files containing data and software. Is the software independent or it has a dependency on having other programs from Nikon? Information on how to use or run it is unclear.

These aspects have been clarified in the revised manuscript.

Figure 1. In the figure legend please indicate if these (B and C) are the awake or urethane anaesthetized mice. Panel B picture 3 of bladder shows significant residual volume, would that be urethane anaesthetized?

This has been indicated in the legend. Example 1 is indeed an anesthetized animal, whereas Example 2 is awake.

Figure 3. In figure legend, indicate that the mice are catheterized.

This has been indicated in figures 2 and 3.

Figure 4. The legend mentions that urethane catheterized mice values taken from figure 2 and 3. However, the residual volume (RV) of urethane dosed mice (catheter, green dot) in figure 4E is < 25 ul, whereas in figure 2D urethane dosed (catheter, blue) mice RV is ~ 60 ul.

There was an erroneous shift in the Y axis for some of the data points taken from figure 2 and 3 in Figure 4. This has been corrected. Thank you for pointing this out!